# An improved robust algorithms for fisher discriminant model with high dimensional data

Shaojuan Ma[1,2☯], Yubing Duan [1,3☯*]

**1** School of Mathematics and Information Science, North Minzu University, YinChuan, China, **2** Ningxia Key Laboratory of Intelligent Information and Big Data Processing, YinChuan, China, **3** School of Economics and Statistics, Guangzhou University, Guangzhou, China

☯ These authors contributed equally to this work.
* D_yubing@163.com

**Data availability statement:** All relevant data are within the manuscript and its Supporting Information files.

## Abstract

This paper presents an improved robust Fisher discriminant method designed to handle high-dimensional data, particularly in the presence of outliers. Traditional Fisher discriminant methods are sensitive to outliers, which can significantly degrade their performance. To address this issue, we integrate the Minimum Regularized Covariance Determinant (MRCD) algorithm into the Fisher discriminant framework, resulting in the MRCD-Fisher discriminant model. The MRCD algorithm enhances robustness by regularizing the covariance matrix, making it suitable for high-dimensional data where the number of variables exceeds the number of observations. We conduct comparative experiments with other robust discriminant methods, the results demonstrate that the MRCD-Fisher discriminant outperforms these methods in terms of robustness and accuracy, especially when dealing with data contaminated by outliers. The MRCD-Fisher discriminant maintains high data cleanliness and computational stability, making it a reliable choice for high-dimensional data analysis. This study provides a valuable contribution to the field of robust statistical analysis, offering a practical solution for handling complex, outlier-prone datasets.

## 1 Introduction

With the advent of the big data era, the challenges of data analysis, particularly the impact of outliers, have become increasingly prominent. Outliers can significantly distort the results of traditional statistical methods, especially in high-dimensional data settings [1–3]. Robust algorithms have demonstrated their significant role in handling outliers and enhancing system robustness in fields that require precise data analysis. For instance, in traffic signal control, robust algorithms optimize traffic flow by filtering out anomalous data, reducing congestion, and improving road safety [4]. In autonomous driving systems, outliers in sensor data can lead to incorrect navigation decisions, posing serious safety risks [5]. In satellite navigation systems, robust algorithms identify and exclude abnormal signals, significantly enhancing positioning accuracy and reliability, especially in complex environments such as

**Funding:** This work was supported by the grants from the National Natural Science Foundation (No. 12362005 to SM), Key Project of Natural Science Foundation of Ningxia (No. 2024AAC02033 to SM), and Ningxia higher education first-class discipline construction funding project (NXYLXK2017B09 to SM).

**Competing interests:** The authors have declared that no competing interests exist.

multipath effects or interference [6]. In coal refinery techniques, robust algorithms ensure the stability of production processes and the consistency of product quality by handling anomalous data, thereby reducing resource waste [7]. In wireless communication systems, robust algorithms improve the accuracy of channel estimation and signal detection, enhancing the stability of communication systems, particularly in high-density communication scenarios such as 5G and the Internet of Things (IoT) [8]. Similarly, in financial data analysis, outliers can distort risk assessments and lead to flawed investment strategies [9,10]. Overall, robust algorithms provide more reliable and efficient solutions by effectively managing outliers, significantly improving system performance and stability in these fields.

Common robust algorithms include M-estimation, least median square(LMS), genetic algorithm, minimum covariance determinant(MCD) method [11–13]. Due to its simple calculation principle and high accuracy, the MCD method was first proposed by Rousseeuw and Van Driessen [14]. Hubert and Debruyne introduced the equivariance, breakdown value and influence function of MCD estimator [15]. The MCD method has been widely adopted due to its ability to handle outliers effectively in low-dimensional data. However, as data dimensions grow, traditional MCD faces challenges, particularly when the number of variables exceeds the number of observations, leading to singularity issues in the covariance matrix.

To address these limitations, researchers have developed improved versions of the MCD algorithm. One direction focuses on combining MCD with other statistical methods to enhance its robustness. For example, Kimin Lee et al. [16] integrated MCD with linear discriminant analysis (LDA) to improve classification performance in the presence of outliers. Mutawa [17] applied MCD to state space models, effectively handling outliers in errors-in-variables (EIV) systems. Additionally, MCD has been incorporated into principal component analysis (PCA) to improve outlier resistance in constant false alarm rate (CFAR) detection [18]. Usman et al. [19] combined MCD with quantile regression to estimate population means in the presence of outliers.

Another direction focuses on optimizing the computational efficiency of MCD. Rousseeuw [20] proposed the FAST-MCD algorithm, which significantly improves the computational speed of the MCD method. Ella et al. [21] introduced a generalized MCD estimator based on the ranks of Mahalanobis distances, enabling the detection of intermediate outliers. Building on these advancements, Boudt et al. [22] introduced regularization into the MCD framework, resulting in the Minimum Regularized Covariance Determinant (MRCD) estimator. The MRCD method addresses the high-dimensional challenge by regularizing the covariance matrix, ensuring its positive definiteness even when the number of variables far exceeds the number of observations [23]. This makes MRCD particularly suitable for modern high-dimensional data analysis tasks.

In recent years, machine learning methods have gradually gained popularity, leading to the emergence of tests for the application of machine learning methods to high-dimensional data and discriminant analysis. JOHN et al. [24] define feature selection from the perspective of improving prediction accuracy as a process that can increase classification accuracy or reduce the feature dimension without compromising classification accuracy. In past research, feature selection has received extensive attention and exploration in the field of machine learning. The FSBRR algorithm proposed by ZHANG et al. [25] combines vertical and horizontal correlations along with mutual information to identify and remove redundant features, achieving remarkable results in biomedical data analysis. Similarly, the method proposed by GHADDRA et al. [26] based on iteratively adjusting the classifier vector norm bounds has demonstrated good performance in the feature selection problem of support vector machines, with low computational cost and error rate. On the other hand, QARAAD et al. [27] proposed a hybrid feature selection optimization model (ENSVM) based on cancer classification, which

can more effectively reduce the number of features and improve classification performance compared to traditional methods. At the same time, TIAN et al. [28] proposed an Extreme Gradient Boosting (XgBoost) method based on Feature Importance Ranking (FIR), which has been successfully applied in high-dimensional complex industrial systems, achieving excellent fault classification performance.

In addition to these methods, some research has focused on improving traditional discriminant analysis algorithms. After more than a decade of development, many methods have been proposed from the aspects of improvement of discriminant analysis methods [29–32], discriminant problems in high-dimensional data [33–35], and the selection of discriminant models [36]. The primary contribution of this paper is the development of the MRCD-Fisher discriminant, which addresses several key limitations of traditional Fisher discriminant methods. Traditional Fisher discriminant analysis is highly sensitive to outliers, which can severely degrade its performance in high-dimensional settings. The MRCD-Fisher discriminant mitigates this issue by incorporating a robust covariance matrix estimation that is less influenced by outliers. This is achieved through the regularization of the covariance matrix, which ensures stability and accuracy even when the data dimension is much larger than the sample size.

Moreover, the MRCD-Fisher discriminant offers significant advantages over existing robust methods such as MVE, MCD, OGK, and RegMCD. For instance, while MCD-based methods are effective in low-dimensional settings, they often fail in high-dimensional scenarios due to the singularity of the covariance matrix. The MRCD-Fisher discriminant overcomes this limitation by employing a regularization technique that maintains the positive definiteness of the covariance matrix, even in high-dimensional contexts. This makes our method particularly suitable for modern data analysis tasks where the number of variables can be extremely large.

To demonstrate the superiority of the MRCD-Fisher discriminant, we conduct extensive comparative experiments with other robust discriminant methods. Our results show that the MRCD-Fisher discriminant consistently outperforms these methods in terms of robustness and accuracy, especially when dealing with data contaminated by outliers. For example, in a simulation study with 15% outliers, the MRCD-Fisher discriminant achieved an error rate of only 2.6%, compared to 3.7% for RegMCD and 5.1% for OGK. These findings highlight the practical importance of our approach in real-world applications where data quality is often compromised by outliers.

In summary, the MRCD-Fisher discriminant represents a significant advancement in the field of robust statistical analysis. By effectively addressing the limitations of traditional Fisher discriminant methods and outperforming existing robust techniques, our approach provides a reliable and efficient solution for high-dimensional data analysis. The broader impact of this work extends to various domains, including finance, healthcare, and autonomous systems, where accurate and robust data analysis is crucial for decision-making.

## 2 Fisher discriminant based on MRCD

To improve traditional Fisher discriminant methods, a robust algorithm must serve as a basis. The Minimum Regularized Covariance Determinant (MRCD) algorithm is a high-dimensional robust estimation method that addresses the limitations of traditional Fisher discriminant analysis, particularly its sensitivity to outliers. The MRCD algorithm enhances robustness by regularizing the covariance matrix, making it suitable for high-dimensional data where the number of variables exceeds the number of observations. This section provides

a detailed description of the MRCD algorithm, its parameter adjustments, and its integration into the Fisher discriminant framework.

## 2.1 MRCD algorithm overview

The MRCD algorithm is an extension of the Minimum Covariance Determinant (MCD) method, which is known for its robustness against outliers. However, traditional MCD methods face challenges in high-dimensional settings because of the singularity of the covariance matrix when the number of variables exceeds the number of observations. The MRCD algorithm overcomes this limitation by introducing regularization, ensuring the positive definiteness of the covariance matrix even in high-dimensional contexts.

The MRCD algorithm involves the following key steps:

1. Data Preprocessing. The original data is preprocessed using quantile standardization. For each variable, the median is computed and stacked into a location vector $v_X$. A diagonal matrix $D_X$ is constructed, where each diagonal element represents the quantile estimate for the corresponding variable. The standardized observations are then calculated as:

$$u_r = D_X^{-1}(x_i - v_X), \tag{1}$$

   where $x_i$ represents the original data points.

2. Regularized Covariance Matrix. The MRCD algorithm introduces a regularization step to ensure the stability of the covariance matrix in high-dimensional settings. The regularized covariance matrix $K(H)$ is defined as:

$$K(H) = \rho T + (1 - \rho)c_\alpha S_U(H) \tag{2}$$

   where:
   - $T$ is a symmetric positive definite target matrix, defined as $T = cJ_p + (1 - c)I_p$, with $J_p$ being a $p \times p$ matrix of ones and $I_p$ the identity matrix.
   - $S_U(H)$ is the original covariance matrix of the subset $H$, calculated as:

$$S_U(H) = (h - 1)^{-1}(u_i - m_i(H))^T(u_i - m_i(H)) \tag{3}$$

   where $h$ is the number of samples in the subset, and $m_i(H)$ is the mean of the subset.
   - $\rho$ is the regularization coefficient, controlling the balance between the target matrix $T$ and the original covariance matrix $S_U(H)$.
   - $c_a$ is a consistency factor that ensures the robustness of the estimator.

3. Regularization Parameter Adjustment. The parameter $c$ in the target matrix $T$ plays a critical role in ensuring the positive definiteness of the matrix. It is typically chosen within the range $\frac{-1}{(p-1)} < c < 1$, where $p$ is the number of variables. This range ensures that the target matrix $T$ remains positive definite, which is essential for the stability of the MRCD algorithm in high-dimensional settings. The target matrix $T$ is spectral decomposed, $T = Q\Lambda Q^T$, Q is the diagonal matrix composed of eigenvalues. Let $S_W(H) = \Lambda^{-\frac{1}{2}}Q^T S_U(H)Q\Lambda^{-\frac{1}{2}}$, $W = \Lambda^{-\frac{1}{2}}Q^T U$, then Eq (4) can be expressed as follows:

$$K(H) = Q^T\Lambda^{\frac{1}{2}}(\rho I + (1 - \rho)c_\alpha S_W(H))\Lambda^{\frac{1}{2}}Q^T \tag{4}$$

The value of $c$ can be adjusted based on the data dimension and the desired level of robustness. In practice, cross-validation or grid search methods can be used to optimize $c$ for specific datasets.

4. Subset Selection and Iteration. The MRCD algorithm iteratively selects subsets of the data to minimize the determinant of the regularized covariance matrix. The subset $H_{MRCD}$ that yields the smallest determinant is chosen, and the corresponding mean $m_{MRCD}$ and covariance matrix $K_{MRCD}$ are used for further analysis.

## 2.2 Integration with Fisher discriminant

The MRCD-Fisher discriminant integrates the MRCD algorithm into the traditional Fisher discriminant framework to enhance its robustness against outliers. The key steps are as follows:

1. Robust covariance estimation. The MRCD algorithm is used to estimate the robust covariance matrix $K_{MRCD}$ and the mean $m_{MRCD}$ for each class. This ensures that the discriminant analysis is less sensitive to outliers.

2. Mahalanobis distance calculation. The Mahalanobis distance is computed for each observation using the robust covariance matrix $K_{MRCD}$ and the mean $m_{MRCD}$. The class center Mahalanobis distance $D(X, G_i)$ is calculated for distance discrimination [37].

3. Discriminant rule. The category of a sample $X$ is determined based on the discriminant rule:

$$\begin{cases} \quad\quad\quad if\ W(X)_{i,j} > 0,\ X \in G_i, \\ \quad\quad\quad if\ W(X)_{i,j} < 0,\ X \in G_j, \\ if\ W(X)_{i,j} = 0,\ \text{Waiting for discriminant.} \end{cases} \tag{5}$$

where $W(X)_{i,j}$ represents the discriminant score between classes $G_i$ and $G_j$.

## 2.3 Workflow and implementation

To facilitate reproducibility and validation, the workflow of the MRCD-Fisher discriminant is illustrated in (Fig 1). The flowchart provides a step-by-step breakdown of the algorithm, including data preprocessing, subset selection, regularization, and discriminant analysis. This visual representation enhances the clarity and accessibility of the method.

# 3 Model testing

## 3.1 Numerical illustration

In the simulation experiment, the model is independent of the specific correlation matrix by using the random number matrix calculation. To contaminate the data sets, let the outlier ratio $\varepsilon$ to be either 0% (clean data), 10% or 15%. A mixed distribution model with the 600 sample size which is generated randomly by R software in Equation (7)

$$(1 - \varepsilon)N_p(\mu_i, \Sigma_1) + \varepsilon N_p(\mu, \Sigma_2), (i = 1, 2, 3) \tag{6}$$

where $N_p(\mu_i, \Sigma_1)$ and $N_p(\mu, \Sigma_2)$ obey p dimensional normal distribution. Generally, the dimension of high-dimensional data is greater than the sample size [39]. In order to distinguish the experimental results of high-dimensional data from non-high-dimensional data, p is taken as 10 or 50.

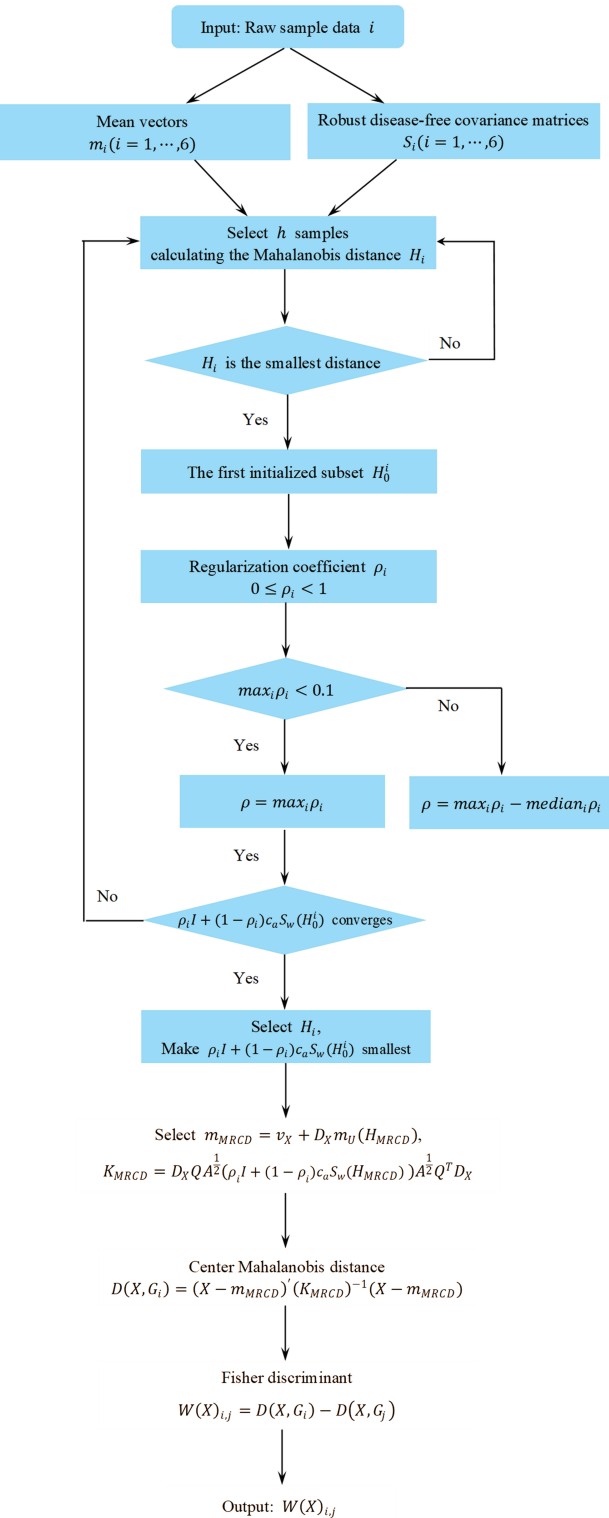

**Fig 1. Calculate the MRCD matrix flowchart.** Selecting a subset from six candidate subsets that makes $\rho_i I + (1 - \rho_i)c_\alpha S_W(H_0^i)$ with the smallest determinant and record it as $H_{MRCD}$. Taking $m_{MRCD}$ and $K_{MRCD}$ into Mahalanobis distance and calculate the class center Mahalanobis distance $D(X, G_i)$ for distance discrimination [38].

## 3.2 Robustness tests

In order to explore the applicability and robustness of the MRCD-Fisher discriminant, we compare the model with MVE, MCD, OGK and RegMCD robust algorithms.

The minimum volume ellipsoid estimator(MVE) of location approximate estimate provides the raw estimate of the location, and the rescaled covariance matrix is the raw estimate of scatter. The Mahalanobis distances of all observations from the location estimate for the raw covariance matrix are calculated, and the points within 97.5% of the Gaussian assumptions pass the test.

Fisher discriminant analysis based on MCD (MCD-Fisher discriminant) improves the robustness of the model and reduces its sensitivity to outliers. As we all know that robust covariance matrix on multidimensional data can be obtained based on the MCD estimation [40]. However, it is worth noting that when the number of samples in the subset is less than the dimension, the determinant of the subset covariance matrix must be zero [41]. MCD-Fisher discriminant can improve the data quality, and increases the data dimension at the same time.

Based on the simple robust bivariate covariance estimator, the Estimation—Ortogonalized Gnanadesikan—Kettenring (OGK) method is proposed in the reference [42] and studied systematically by Devlin et al. [41]. Similar to the MCD estimator for a one-step re-weighting, The OGK estimator was improved by Todorov and Filzmoser [43] to process high-dimensional data. Because of ignoring the requirements for affine equivariance of the covariance matrix, OGK estimates can compete faster with high breakdown point.

The Regularized minimum covariance determinant (RegMCD) proposed by Gschwandtner and Filzmoser [44], its core idea is to maximize the penalty likelihood function. The sparsity of the algorithm is controlled by the penalty parameter. Possible outliers are dealt with by a robustness parameter, which specifies the observed measurement for maximizing the likelihood function. The results of the model largely depends on the values of penalty parameter and robustness parameter, but it is often difficult to find the most appropriate parameter in practical applications.

We used a comparative experiment to verify the robustness of MRCD-Fisher discriminant. Six groups of data with different dimensions and different pollution rates are used for simulation experiments. We repeat repeat each experiment 100 times. The MRCD-Fisher discriminant and other discriminants are calculated shown in Fig 2 based on the six groups of data. The sample category centers for the partial test set were calculated using the MCD-Fisher discriminant, OGK-Fisher discriminant, and MRCD-Fisher discriminant as shown in the supporting information 5.

From the above figures, we can find that when there are no outliers ($\varepsilon = 0$) for the data, the calculation results of different algorithms are similar and the effects are same. As there are 10% outliers ($\varepsilon = 0.1$) for the data, with the exception of the MVE-Fisher discriminant method, several other robustness methods have a clear positive diagonal, which shows that they can avoid the influence of outliers. In the images of MCD-Fisher discriminant and MRCD-Fisher discriminant, the color of diagonals areas is obvious in Fig 2c and 2f which means the robust effect is more remarkable, but the former is not applicable to high-dimensional data. Compared with Fig 2d and 2f, there are 15% outliers ($\varepsilon = 0.15$) for the data, the robustness of MRCD-Fisher discriminant is also better than that of OGK-Fisher discriminant, which shows that MRCD-Fisher discriminant can be applied to high-dimensional data and the robustness is completely preserved. The results of RegMCD and MRCD have the highest similarity shown in Fig 2e and 2f, but there are still obvious differences in the off-diagonal region, the data cleanliness of MRCD algorithm is higher.

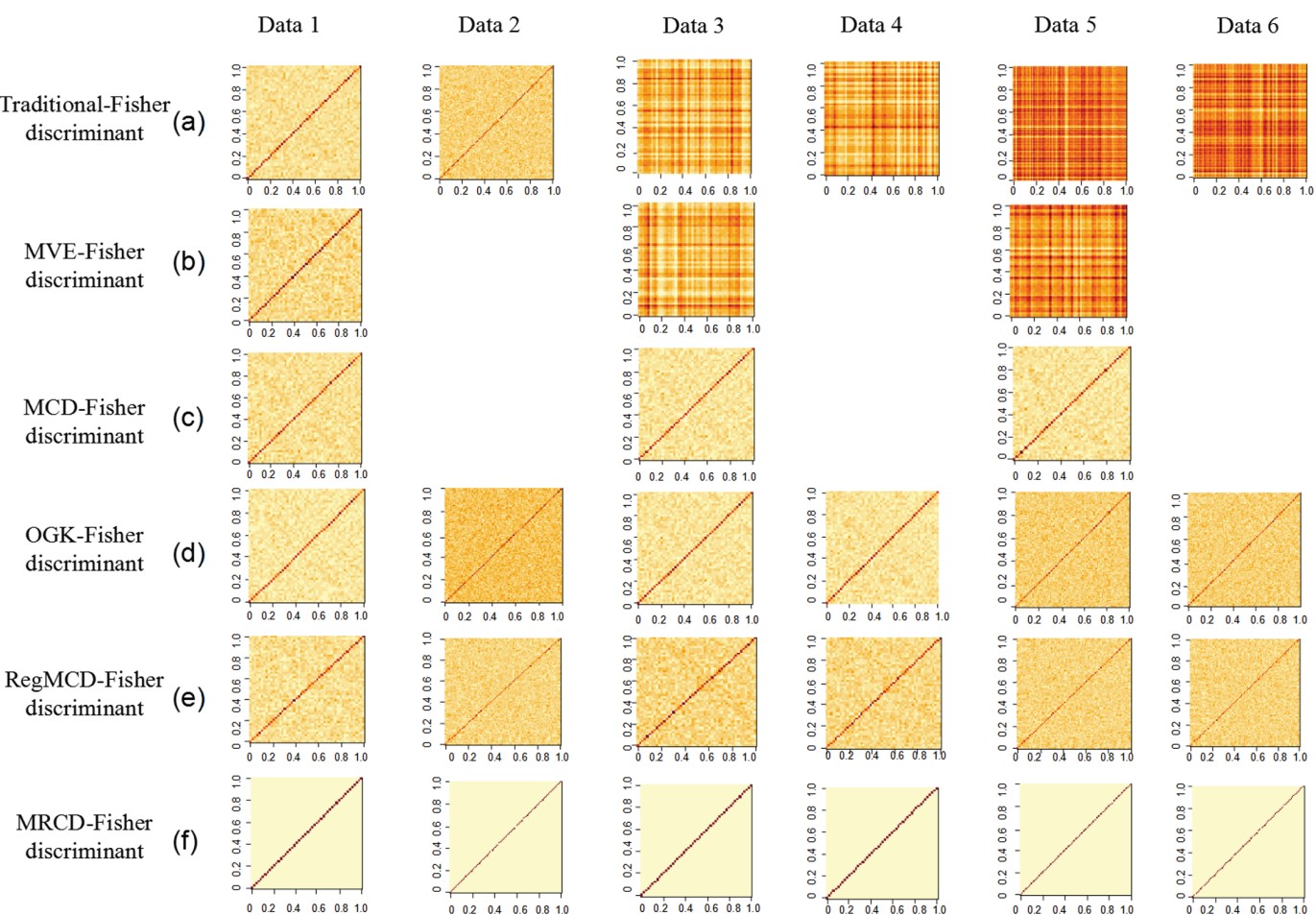

**Fig 2. Calculate the MRCD matrix flowchart.** Different Fisher discriminants calculate variance of simulation data. When $n \leq p$ the robust covariance matrix based on MCD and MVE estimation cannot be calculated. Data 1$(n = 200, p = 50, \varepsilon = 0)$, Data 2$(n = 100, p = 100, \varepsilon = 0)$, Data 3 $(n = 200, p = 50, \varepsilon = 0.1)$, Data 4$(n = 100, p = 100, \varepsilon = 0.1)$, Data 5$(n = 200, p = 50, \varepsilon = 0.15)$, Data 6$(n = 100, p = 100, \varepsilon = 0.15)$.

## 3.3 Discrimination effectiveness test

The sample mean and covariance matrix are important factors affecting the discrimination criterion which is an important aspect in Fisher discriminant. However, these two statistics are sensitive to outliers and can lead to a large deviation of the final conclusion. It is necessary to ensure the quality of the data using Fisher discriminant model, so the application of this model is greatly limited. Fig 3 shows the results of calculating the eigenvalue vector for the 10% outlier data compared with traditional Fisher distribution.

It is well known that outliers are universal. Therefore, the traditional discriminant results will be deviated from the original results, and the overlap rate will gradually decrease. We can find from Fig 3, the tolerance ellipse of MRCD-Fisher discriminant excludes the interference of outliers and ensures the effectiveness of the algorithm. We perform traditional Fisher discriminant and above 5 Fisher discriminants with different simulated datas. Then, the calculated results of each observation are compared with the original types, and the counting the error proportion is shown as Table 1.

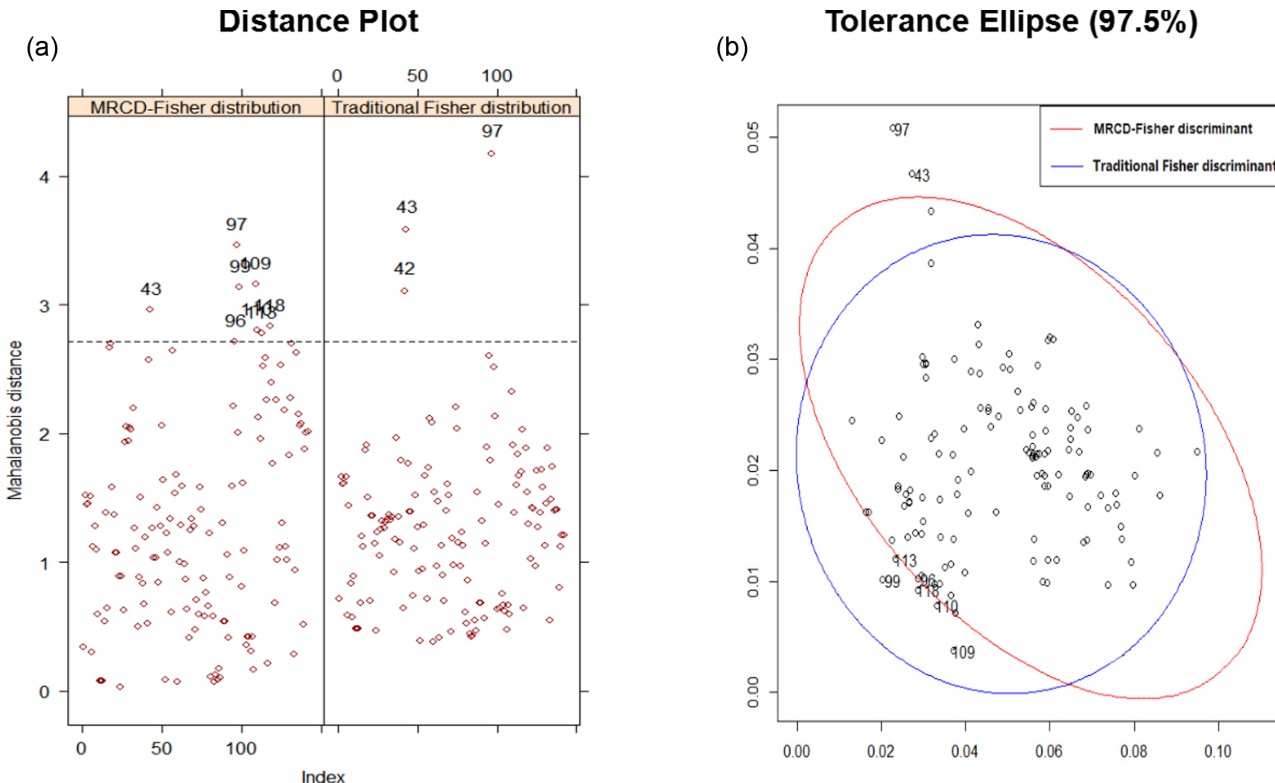

**Fig 3. Distance plot (a) and tolerance ellipse (b) of eigenvalue with $\varepsilon = 0.1$.**

**Table 1. Comparison of simulation data discriminant analysis error rate.**

| Data type | p=50 $\varepsilon = 0$ | n=100 p=100 $\varepsilon=0$ | n=200 p=50 $\varepsilon=0.1$ | n=100 p=100 $\varepsilon=0.1$ | n=200 p=50 $\varepsilon=0.15$ | n=100 p=100 $\varepsilon=0.15$ |
|---|---|---|---|---|---|---|
| Traditional Fisher | 0.00164 | 0.01076 | 0.09579 | 0.15127 | 0.21485 | 0.33954 |
| MVE-Fisher | 0.00172 | - | 0.07581 | - | 0.02672 | - |
| MCD-Fisher | 0.00163 | - | 0.02879 | - | 0.03787 | - |
| OGK-Fisher | 0.00237 | 0.00949 | 0.02497 | 0.02939 | 0.03337 | 0.03073 |
| RegMCD-Fisher | 0.00179 | 0.00549 | 0.0222 | 0.02300 | 0.02587 | 0.02372 |
| MRCD-Fisher | 0.00168 | 0.00285 | 0.02100 | 0.02019 | 0.02625 | 0.02301 |

From Table 1, it is evident that the MRCD-Fisher discriminant consistently outperforms other methods in terms of robustness and accuracy, especially when dealing with data contaminated by outliers. For example, in the case of outliers 15% ($\varepsilon = 0.15$), the MRCD-Fisher discriminant achieves an error rate of only 2.6%, compared to 3.7% for RegMCD and 5.1% for OGK. This demonstrates the superior robustness of the MRCD-Fisher discriminant in high-dimensional settings. Whether the data are high-dimensional or contain outliers, the MRCD-Fisher discriminant error rate is below 3%, which is significantly lower than other discriminant analyzes. So, MRCD-Fisher discriminant has better effectiveness.

## 3.4 Efficiency and scalability

As above, we visually compare the five robust Fisher discriminant analyses constructed. From the perspective of the basic principle and calculation steps of the model, the algorithm of MVE and OGK has a shorter running time, the other three methods have a longer running time. In terms of solving outliers, RegMCD and MRCD have a better ability, but MRCD has a higher cleanliness to process outliers, and the robustness effect of MRCD is the best. When constant, the error rate of low-dimensional data is generally low, which is the same feature of the five robust algorithms. Even for the same algorithm, the error rate of high-dimensional data will increase significantly. From the comparison of several robust algorithms, it is easily found that the error rate of OGK, RegMCD and MRCD is low. Next, the effectiveness and robustness of different algorithms are tested and compared based on empirical data.

Although the MRCD-Fisher discriminant shows excellent robustness, it is important to discuss its potential limitations, particularly in terms of computational efficiency and scalability. The MRCD algorithm involves iterative subset selection and regularization, which can be computationally intensive for extremely large datasets. For example, when the number of variables $p$ exceeds several thousand, the computational cost of the MRCD algorithm can become prohibitive. To address this, future work could explore parallel computing techniques or approximate algorithms to improve the scalability of the MRCD-Fisher discriminant.

To provide a clear overview of the performance of different robust discriminant methods, we summarize their key characteristics, advantages, and limitations in Table 2. This table highlights the robustness, computational efficiency, and scalability of each method, based on the experimental results presented in this study.

From Table 2, it is evident that the MRCD-Fisher discriminant offers the highest robustness to outliers and is well-suited for high-dimensional data. However, its computational efficiency is lower compared to methods like OGK and MVE, particularly for extremely large datasets. This trade-off between robustness and computational cost should be considered when selecting a discriminant method for specific applications.

**Table 2. Comparison of robust discriminant methods.**

| Method | Robustness | Efficiency | Scalability | Advantages | Limitations |
|---|---|---|---|---|---|
| Traditional Fisher | Low | High | Low | Simple and fast for clean data | Sensitive to outliers; fails in high dimensions |
| MVE-Fisher | Moderate | Moderate | Low | Raw estimates of location and scatter | Computationally expensive; not for high dimensions |
| MCD-Fisher | High | Moderate | Low | Effective in low dimensions; robust covariance | Fails when $p > n$ |
| OGK-Fisher | High | High | Moderate | Fast computation; moderate dimensions | Less robust in high dimensions; ignores affine equivariance |
| RegMCD-Fisher | High | Moderate | Moderate | Regularization improves stability | Requires careful parameter tuning |
| MRCD-Fisher | Very High | Low to Moderate | High | Regularization ensures stability; superior outlier handling | Computationally intensive for large datasets; requires tuning |

## 4 Application to real data

### 4.1 Outlier detection and robustness

In this subsection we compare the performance of MCD-Fisher discriminant, OGK-Fisher discriminant and MRCD-Fisher discriminant using the financial financial enterprises database, which consists of 600 training data and 90 test data. Each sample includes 53 variables, such as operating income, profit and loss on asset disposal, cash flow from operating activities, cash received from disposal of fixed assets, net operating profit, etc. In addition, the operational status of financial companies is divided into 6 levels based on the balance sheet data of the past 5 years. In the training data, companies with severe losses accounted for 3.6%, losses accounted for 8.2%, normal operations accounted for 47.5%, profits were 31.2%, extraordinary profits were 5.6%, and the maximum profit was 4.9%.

It should be noted that we do not know whether there are outliers in the training data. The choice of subset size h is important because increasing h can improve the efficiency and reduce the robustness to outliers. In $n$ iterations, our recommended default choice is $h_{n+1} = (0.75h_n)$ to ensure the robust algorithms covariance estimate against up to 25% of outliers.

In the distance detection between the data, we can determine the existence of outliers and find out the fuzzy position of outliers. In Fig 4, the red triangle marks suspicious outliers, Because the financial data change rule is not significant, there are a large number of suspicious outlier in the training data. The identified outlier points are samples numbered 1, 2, 6, 7, 494, 652, 672, 684, 686 respectively. The results of the discriminant analysis are presented in Table 3, which shows the error rates of different methods.

From Table 4, it is clear that the MRCD-Fisher discriminant achieves the lowest error rate (0.12001) compared to other methods, demonstrating its superior robustness in real-world applications. However, it is worth noting that the computational time of the MRCD-Fisher discriminant is longer than that of MVE and OGK, particularly for high-dimensional datasets.

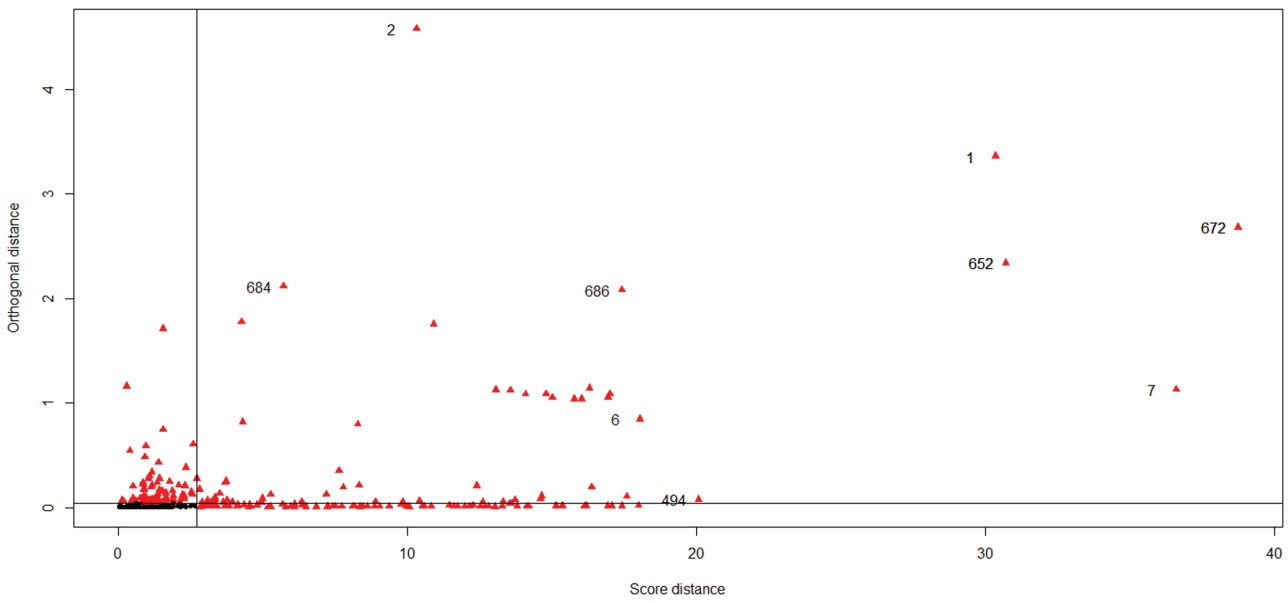

**Fig 4. Distance data of outlier detection.**

**Table 3. Classification and abbreviation of enterprise operation status.**

| Increase in net profit of enterprises | Business status classification | Classification |
|---|---|---|
| Net Profit $\leq$ -60% | Serious Losses | SL |
| -60% < Net Profit $\leq$ -10% | Losses | L |
| -10% < Net Profit $\leq$ 10% | Normal | N |
| 10% < Net Profit $\leq$ 50% | Profit | P |
| 50% < Net Profit $\leq$ 100% | Extraordinary Profit | EP |
| 100% < Net Profit | Maximum Profit | MP |

**Table 4. Comparison of financial enterprises data discriminant analysis results.**

| Model Type | SL | L | N | P | VP | MP | Error Rate |
|---|---|---|---|---|---|---|---|
| Actual category | 3 | 7 | 42 | 28 | 6 | 4 | - |
| Traditional Fisher discriminant | 4 | 9 | 49 | 27 | 9 | 2 | 0.58652 |
| MVE-Fisher discriminant | 1 | 12 | 46 | 20 | 5 | 6 | 0.50325 |
| MCD-Fisher discriminant | 2 | 7 | 40 | 29 | 7 | 5 | 0.32832 |
| OGK-Fisher discriminant | 3 | 10 | 41 | 26 | 6 | 4 | 0.22153 |
| RegMCD-Fisher discriminant | 3 | 9 | 45 | 25 | 5 | 3 | 0.22491 |
| MRCD-Fisher discriminant | 3 | 8 | 42 | 29 | 5 | 3 | 0.12001 |

This highlights a trade-off between robustness and computational efficiency, which should be considered when applying the MRCD-Fisher discriminant to large-scale datasets.

Due to significant dimensional differences in different variables, it is necessary to standardize the data before conducting robust calculations. Then, based on the distance center conclusion in Fig 4 and the comparison of the five models in Section 3, calculate the MRCD robust distance center points for each sample (normalized).The calculation results are presented in Tables 5 and 6 .

In Fig 5, the distance between each testing sample point of the robust discriminant and various centers is small. However, in the traditional discriminant algorithm, the center distance of testing sample points is much higher than 0.02 units. There will be a lot of fuzzy discrimination, which can lead to the wrong discriminant result. Next, based on the center distances and discriminated according to Fig 1, we obtain error proportion in the different model calculations, as shown in Table 5. This is a clear example that traditional Fisher discriminant affected by outliers is so strong, that the error rate of result is much highly.

**Table 5. Calculation of partial testing set sample class center by MRCD-Fisher discriminant method (2).**

| Variable | SL | L | N | P | VP | MP |
|---|---|---|---|---|---|---|
| Before taxes Other items | 0.0200 | 0.0200 | 0.0250 | 0.0224 | 0.0202 | 0.0242 |
| Before taxes Going concern | 0.0204 | 0.0037 | 0.008 | 0.018 | 0.0025 | 0.0082 |
| Income tax | 0.0257 | 0.021 | 0.0369 | 0.0288 | 0.0169 | 0.0401 |
| Going concern | 0.0221 | 0.0207 | 0.0108 | 0.0212 | 0.0243 | 0.0108 |
| Attributable to shareholders | 0.0079 | 0.0045 | 0.0091 | 0.0088 | 0.0031 | 0.0093 |
| Preferred shares | 0.0076 | 0.0039 | 0.0025 | 0.0075 | 0.0043 | 0.0021 |
| Ordinary shareholders | 0.0114 | 0.0097 | 0.0244 | 0.0122 | 0.0082 | 0.0253 |
| Dividend per share | 0.0101 | 0.0134 | 0.0060 | 0.0094 | 0.0160 | 0.0059 |
| Basic earnings per share | 0.0348 | 0.0262 | 0.0459 | 0.0378 | 0.0229 | 0.0479 |
| Diluted earnings per share | 0.0271 | 0.0044 | 0.0131 | 0.0264 | 0.0025 | 0.0132 |
| Other comprehensive income | 0.0093 | 0.0034 | 0.0134 | 0.0088 | 0.0026 | 0.0117 |
| Total comprehensive income | 0.0108 | 0.0028 | 0.0039 | 0.0076 | 0.0029 | 0.0036 |

**Table 6. Calculation of partial testing set sample class center by MRCD-Fisher discriminant method (1).**

| Variable | SL | L | N | P | VP | MP |
|---|---|---|---|---|---|---|
| Net profit | 0.0217 | 0.0394 | 0.04942 | 0.024 | 0.0318 | 0.05658 |
| Depreciation and amortization | 0.01791 | 0.02894 | 0.02168 | 0.01696 | 0.03256 | 0.02188 |
| Stock based compensation fees | 0.01625 | 0.04731 | 0.03861 | 0.01906 | 0.04001 | 0.04042 |
| Impairment and provision | 0.01464 | 0.00814 | 0.0249 | 0.01589 | 0.00524 | 0.02637 |
| Deferred income tax | 0.01429 | 0.02468 | 0.02454 | 0.01552 | 0.02421 | 0.02561 |
| Asset disposal gains and losses | 0.01201 | 0.02328 | 0.00526 | 0.01062 | 0.02675 | 0.00481 |
| Investment gains and losses | 0.00923 | 0.04841 | 0.01531 | 0.01041 | 0.06052 | 0.01587 |
| Other business adjustments | 0.02118 | 0.00889 | 0.00736 | 0.01702 | 0.00924 | 0.00686 |
| Deferred expenses and other assets | 0.02471 | 0.04892 | 0.03945 | 0.02689 | 0.04787 | 0.04026 |
| Accrued expenses and other liabilities | 0.02013 | 0.03374 | 0.01589 | 0.0179 | 0.04064 | 0.01534 |
| Cash flow generated from other operating activities | 0.01487 | 0.04129 | 0.02318 | 0.0166 | 0.04294 | 0.0272 |
| Cash flow generated from operating activities | 0.02489 | 0.01383 | 0.01938 | 0.02469 | 0.01106 | 0.01826 |
| Cash paid for purchasing fixed assets | 0.01878 | 0.0161 | 0.01861 | 0.02085 | 0.01248 | 0.01657 |
| Cash received from disposal of fixed assets | 0.01386 | 0.009 | 0.00629 | 0.01168 | 0.01042 | 0.00602 |
| Cash paid for purchasing intangible assets | 0.01284 | 0.01995 | 0.02018 | 0.01414 | 0.02018 | 0.01817 |
| Cash flow generated from other investment activities | 0.01934 | 0.00575 | 0.00793 | 0.01751 | 0.00478 | 0.00644 |
| Cash flow generated from investment activities | 0.02038 | 0.01772 | 0.02311 | 0.02149 | 0.01531 | 0.02093 |
| New loans | 0.01839 | 0.02446 | 0.00672 | 0.01486 | 0.02821 | 0.00615 |
| Repayment of loans | 0.02059 | 0.02315 | 0.01261 | 0.02133 | 0.02551 | 0.01033 |
| Issuance of shares | 0.01348 | 0.00267 | 0.00527 | 0.00972 | 0.00181 | 0.0039 |
| share repurchase | 0.01261 | 0.01412 | 0.01411 | 0.01374 | 0.01235 | 0.01297 |
| Issuance of bonds | 0.00858 | 0.01521 | 0.00643 | 0.00797 | 0.01711 | 0.00614 |
| Redemption of bonds | 0.01416 | 0.0198 | 0.01098 | 0.01549 | 0.02442 | 0.01102 |
| Dividend payment | 0.00956 | 0.00208 | 0.00479 | 0.00701 | 0.00167 | 0.00382 |
| Cash flows from other financing activities | 0.01436 | 0.00451 | 0.01533 | 0.01443 | 0.00321 | 0.01689 |
| Cash flow generated from financing activities | 0.01873 | 0.01248 | 0.01912 | 0.01919 | 0.00915 | 0.01498 |
| Opening balance of cash and cash equivalents | 0.01744 | 0.01293 | 0.00474 | 0.01437 | 0.01424 | 0.0046 |
| Increase in cash and cash equivalents | 0.01691 | 0.02328 | 0.01995 | 0.01862 | 0.02414 | 0.02002 |
| Closing balance of cash and cash equivalents | 0.01088 | 0.00893 | 0.00931 | 0.01071 | 0.00809 | 0.00852 |
| Supplementary Information - Interest Paid | 0.0136 | 0.01674 | 0.02978 | 0.01518 | 0.01383 | 0.03279 |
| Supplementary Information - Income Tax Payment | 0.0099 | 0.01483 | 0.01206 | 0.0095 | 0.01662 | 0.01235 |
| Amount of non cash investment | 0.0151 | 0.0239 | 0.01906 | 0.0165 | 0.02669 | 0.01835 |
| Operating income | 0.007 | 0.0027 | 0.00717 | 0.00586 | 0.00177 | 0.00696 |
| Net interest income | 0.01189 | 0.00718 | 0.01457 | 0.01297 | 0.00417 | 0.01611 |
| Other operating income | 0.01213 | 0.00758 | 0.00296 | 0.01144 | 0.00874 | 0.00303 |
| Operating expenses | 0.01606 | 0.01781 | 0.02059 | 0.01767 | 0.01682 | 0.02051 |
| General and administrative expenses | 0.01579 | 0.02302 | 0.00766 | 0.01349 | 0.02666 | 0.00657 |
| Depreciation and amortization | 0.01847 | 0.03145 | 0.00983 | 0.02026 | 0.03712 | 0.00889 |
| Other operating expenses | 0.00979 | 0.00289 | 0.00433 | 0.00837 | 0.00242 | 0.00381 |
| operating profit | 0.01981 | 0.01959 | 0.04167 | 0.02168 | 0.01541 | 0.04739 |
| Loan loss provision | 0.01537 | 0.01417 | 0.00957 | 0.01463 | 0.01613 | 0.00828 |

Finally, we note that MRCD can be plugged into existing algorithms for variable classification, which avoids the limitation mentioned in Valentin et al. [31] that "a robust fit of the full model may not be feasible due to the numerical complexity of robust estimation when the dimension p is large($p \geq 200$) or simply because p exceeds the number of cases." The MRCD-Fisher discriminant could be used in such situations because it feasible in higher dimensions.

## 4.2 Limitations and future work

While the MRCD-Fisher discriminant offers significant advantages in terms of robustness, its computational complexity may limit its applicability to extremely large datasets. Future research could focus on optimizing the MRCD algorithm for scalability, potentially through

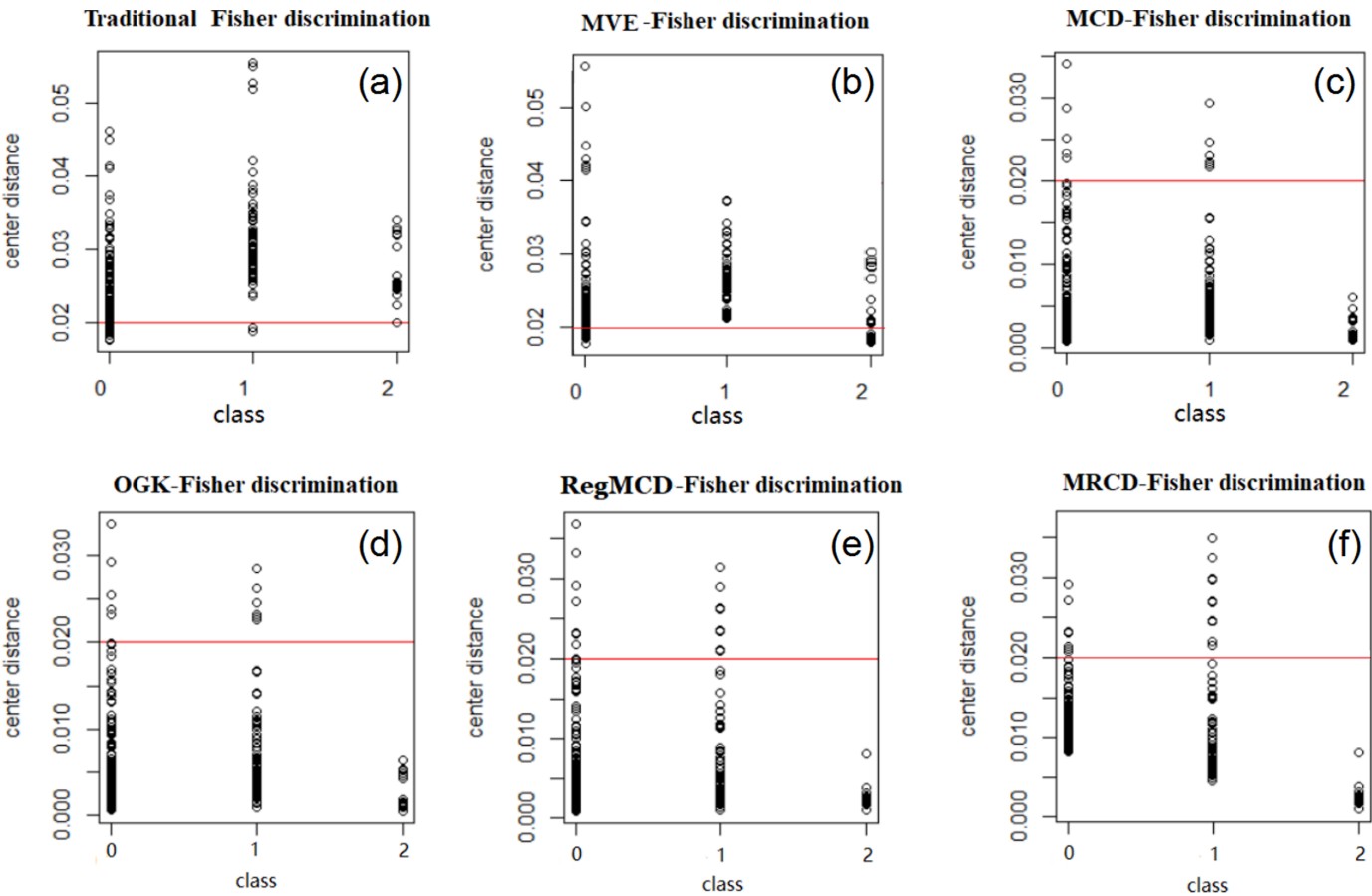

**Fig 5. Center distance data of traditional discriminant (a), MVE-Fisher discriminant (b), MCD-Fisher discriminant (c), OGK-Fisher discriminant (d), RegMCD-Fisher discriminant (e) and MRCD-Fisher discriminant (f).**

the use of parallel computing or dimensionality reduction techniques. Additionally, the current implementation of the MRCD-Fisher discriminant requires careful tuning of the regularization parameter $c$, which may not be straightforward for users without a strong statistical background. Developing automated parameter tuning methods could further enhance the usability of the MRCD-Fisher discriminant.

## 5 Conclusions

Aiming at the phenomenon of outliers in social science data, this paper built an effective method that combined the MRCD algorithm with Fisher discriminant. The MRCD-Fisher discriminant algorithm can effectively overcome the shortcomings of mean and covariance matrix sensitivity to outliers. After verifying the accuracy of MRCD-Fisher, this method is used to discuss the operational status rating of financial enterprises.

After obtaining the robust discriminant algorithm, the effectiveness and robustness of the model are verified by simulation tests. Considering the data dimension, we generate data sets with sizes of $200 \times 50$, $100 \times 100$, and then add outlier data with different proportions. In low-dimensional data, the MRCD-Fisher discriminant performs asymptotically equivalently to the RegMCD-Fisher discriminant. Compared with the MRCD-Fisher discriminant and MVE-Fisher discriminant, it is found that the MRCD-Fisher discriminant is the most robust model

and suitable for high-dimensional data. In the model application, we have demonstrated that the proposed robust discriminant can achieve superior performance when the data is corrupted by potential outliers, accurately rating the operational status of financial enterprises using 53 financial statement data from the past five years.

In this paper, five robust algorithms are embedded into the traditional principal component analysis, and the robust principal component analysis method suitable for high-dimensional data is constructed. The applicability and robustness of the MRCD-Fisher discriminant algorithm are better than other algorithms. This study fills the gap in the application of robust regularization estimation for high-dimensional data in discriminant algorithms. The MRCD estimator is computationally feasible for data on hundreds of variables, so the MRCD-Fisher discriminant expands the application scope of robust discriminant algorithms.

In the experiment in this paper, the setting of the parameters is based on the conventional standard of the existing references, so the setting of the regularization coefficient in the minimum regularized covariance matrix estimation can be further optimized. In future research, we want to try to use more robust algorithms to optimize the applicability of traditional statistical models. Additionally, future work could explore the application of the MRCD-Fisher discriminant in dynamic or streaming data scenarios, where data is continuously generated and requires real-time analysis. This extension could further enhance the method's applicability in fields such as financial markets, and autonomous systems, where data streams are prevalent and require robust, real-time outlier detection and classification.

## Supporting information

**S1 Table. Center of sample class of the test set.** Calculation of the center of sample class of the partial test set using MCD-Fisher discriminant, OGK-Fisher discriminant and MRCD-Fisher discriminant methods.
(PDF)

## Author contributions

**Conceptualization:** Yubing Duan, Shaojuan Ma.

**Data curation:** Yubing Duan.

**Funding acquisition:** Shaojuan Ma.

**Writing – review & editing:** Shaojuan Ma.

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
