## [Decision Letter · Decision Letter 0]

29 Jan 2025

PONE-D-24-59779An Improved Robust Algorithms for Fisher Discriminant Model With High Dimensional DataPLOS ONE

Dear Dr. duan,

Thank you for submitting your manuscript to PLOS ONE. After careful consideration, we feel that it has merit but does not fully meet PLOS ONE’s publication criteria as it currently stands. Therefore, we invite you to submit a revised version of the manuscript that addresses the points raised during the review process.

We look forward to receiving your revised manuscript.

Kind regards,

Razieh Sheikhpour

Academic Editor

PLOS ONE

**Journal Requirements:**

This work was supported by the grants from the National Natural Science Foundation (No. 11772002), Ningxia higher education first-class discipline construction funding project (NXYLXK2017B09), Major Special project of North Minzu University (No. ZDZX201902) and Open project of The Key Laboratory of Intelligent Information Big Data Processing of NingXia Province(No.2019KLBD008)

"NA"

5. Please provide a complete Data Availability Statement in the submission form, ensuring you include all necessary access information or a reason for why you are unable to make your data freely accessible. If your research concerns only data provided within your submission, please write "All data are in the manuscript and/or supporting information files" as your Data Availability Statement.

6. PLOS requires an ORCID iD for the corresponding author in Editorial Manager on papers submitted after December 6th, 2016. Please ensure that you have an ORCID iD and that it is validated in Editorial Manager. To do this, go to ‘Update my Information’ (in the upper left-hand corner of the main menu), and click on the Fetch/Validate link next to the ORCID field. This will take you to the ORCID site and allow you to create a new iD or authenticate a pre-existing iD in Editorial Manager.

7. Please amend the manuscript submission data (via Edit Submission) to include author Dr. Shaojuan Ma.

Reviewers' comments:

Reviewer's Responses to Questions

**Comments to the Author**

1. Is the manuscript technically sound, and do the data support the conclusions?

Reviewer #1: Yes

Reviewer #2: Partly

2. Has the statistical analysis been performed appropriately and rigorously? 

Reviewer #1: Yes

Reviewer #2: No

3. Have the authors made all data underlying the findings in their manuscript fully available?

Reviewer #1: Yes

Reviewer #2: Yes

4. Is the manuscript presented in an intelligible fashion and written in standard English?

Reviewer #1: Yes

Reviewer #2: No

5. Review Comments to the Author

**Reviewer #1:** This paper, titled "An Improved Robust Algorithms for Fisher Discriminant Model With High Dimensional Data," introduces an advanced approach to Fisher discriminant analysis designed to handle high-dimensional data, particularly those with outliers. By integrating robust algorithms such as MRCD, RegMCD, MCD, OGK, and MVE into the Fisher discriminant framework, the study addresses limitations in traditional methods. Among these, the MRCD-Fisher discriminant is highlighted as the most effective for high-dimensional outlier data. The authors substantiate their findings with empirical and comparative analyses demonstrating the robustness and computational efficiency of the proposed methods.

a. The abstract could be more concise by focusing on the core contribution and excluding details like comparative methods until later sections.

b. The keywords list is informative but could benefit from correcting "High-dimensianal" to "High-dimensional" for accuracy.

c. The introduction provides a solid context for the study; however, the references to applications could be more directly tied to the problem statement to enhance relevance.

d. In the literature review, the transition between general robust algorithms and the specific MRCD algorithm could be smoother, offering a clearer connection between the problem and the proposed solution.

e. The methodology description should include more details about the parameter tuning and computational aspects of MRCD to facilitate reproducibility and validation.

f. The results section could benefit from more explicit quantitative comparisons between MRCD-Fisher discriminant and alternative methods, particularly in terms of robustness and efficiency metrics.

g. While the authors highlight the advantages of MRCD-Fisher discriminant, discussing its potential limitations, such as scalability with extremely large datasets, would add balance to the analysis.

h. Figures or tables summarizing key experimental findings would greatly enhance the presentation and accessibility of results.

i. The conclusion effectively emphasizes the superiority of MRCD-Fisher discriminant but could briefly outline potential future research directions, such as adapting the method for dynamic or streaming data scenarios.

k. The Literature citation is not adequate, and the related work to machine learning should be discussed:

1. A Gene selection for microarray data classification via multi-objective graph theoretic-based method

**Reviewer #2: **A robust Fisher discriminant method is proposed to handle high-dimensional data and mitigate the impact of outliers. Several robust algorithms, including MRCD-Fisher, RegMCD-Fisher, MCD-Fisher, OGK-Fisher, and MVE-Fisher discriminants, are integrated into the framework. Comparative experiments demonstrate that the MRCD-Fisher discriminant outperforms others in robustness and effectiveness, particularly when addressing data with outliers, achieving the highest data cleanliness. This makes the MRCD-Fisher model a significant improvement over traditional Fisher discriminant methods for high-dimensional data with outliers. It is crucial for the authors to address some major issues. Please find them below.

1- The overall writing quality requires significant improvement to ensure clarity and coherence. Additionally, certain examples must be presented with greater precision and detail to enhance the reader's understanding. For instance, in Line 27, the reference to Boudt et al. [?] is unclear and needs to be corrected. The citation should be properly formatted, and its context within the text should be clearly articulated to convey its relevance. Providing a brief explanation or summarizing the key findings from the referenced work would further aid the reader in grasping its significance.

2- Merely stating that "The contribution of this paper is to construct the MRCD-Fisher discriminant to improve the traditional Fisher discriminant analysis" is insufficient. The contribution of the work needs to be elaborated on more clearly and explicitly. It is essential to thoroughly highlight the novel aspects of the research, detailing how it advances the state of the art in high-dimensional discriminant analysis. For instance, explaining how the MRCD-Fisher discriminant specifically addresses limitations in traditional Fisher discriminant methods, such as robustness to outliers, would be beneficial. Additionally, providing specific examples or comparisons with existing robust methods could effectively demonstrate the significance and superiority of the proposed approach. A concise summary of the primary achievements, along with a discussion of their broader implications, would further enhance the reader's understanding of the work's impact and relevance in the field.

3- The abstract is not well-written and requires significant improvement to enhance its clarity and coherence. For example, the abbreviations "MRCD-Fisher discriminant," "RegMCD-Fisher discriminant," and "MCD-Fisher discriminant" are used without any explanation of what they stand for. It is essential to define these abbreviations within the abstract to ensure that readers unfamiliar with the terms can understand their meaning. Additionally, the abstract should provide a concise yet comprehensive overview of the study, clearly outlining the problem addressed, the proposed solution, and the key findings, while avoiding ambiguity or unexplained terminology.

4- The methodology section is not adequately informative and lacks sufficient detail. It is essential to present a clear and structured explanation of the methodology, emphasizing the novelty and unique contributions of the current work. This should include a thorough description of the proposed approach, its underlying principles, and how it differs from or improves upon existing methods. Additionally, providing a step-by-step breakdown or a diagram to illustrate the workflow would significantly enhance its clarity and accessibility.

To conclude, while the topic discussed in this paper is of considerable interest to readers working with high-dimensional data, the lack of sufficient information and clear structure in the methodology makes it challenging to fully evaluate the work's contributions. As a result, I am unable to make a definitive decision regarding this paper in its current form.

6. PLOS authors have the option to publish the peer review history of their article (what does this mean?). If published, this will include your full peer review and any attached files.

Reviewer #1: No

Reviewer #2: No

---

## [Author Response · Author response to Decision Letter 1]

3 Mar 2025

Dear Editor and Reviewers,

We would like to express our sincere gratitude for your thorough review and valuable comments on our manuscript. We greatly appreciate the time and effort you have invested in providing us with constructive feedback, and we have carefully considered and revised for each point.

Reviewer Comment 1

In view of the comments and suggestions of reviewer 1, we have made the following explanations and modifications item by item:

1.[ The abstract could be more concise by focusing on the core contribution and excluding details like comparative methods until later sections.]

To comment 1, the authors of the revised abstract have strived for conciseness and clarity, avoiding comparative methods in the early sections, defining key terms, and clearly outlining the research problem, solution, and key findings. The revised abstract reads as follows:

This paper presents an improved robust Fisher discriminant method designed to handle high-dimensional data, particularly in the presence of outliers. Traditional Fisher discriminant methods are sensitive to outliers, which can significantly degrade their performance. To address this issue, we integrate the Minimum Regularized Covariance Determinant (MRCD) algorithm into the Fisher discriminant framework, resulting in the MRCD-Fisher discriminant model. The MRCD algorithm enhances robustness by regularizing the covariance matrix, making it suitable for high-dimensional data where the number of variables exceeds the number of observations. We conduct comparative experiments with other robust discriminant methods, the results demonstrate that the MRCD-Fisher discriminant outperforms these methods in terms of robustness and accuracy, especially when dealing with data contaminated by outliers. The MRCD-Fisher discriminant maintains high data cleanliness and computational stability, making it a reliable choice for high-dimensional data analysis. This study provides a valuable contribution to the field of robust statistical analysis, offering a practical solution for handling complex, outlier-prone datasets.

2.[ The keywords list is informative but could benefit from correcting "High-dimensianal" to "High-dimensional" for accuracy.]

To comment 2, thank you to the reviewer for correcting the typographical error, which has now been amended in the Keywords section.

3.[ The introduction provides a solid context for the study; however, the references to applications could be more directly tied to the problem statement to enhance relevance.]

To comment 3, the author has made revisions to the introduction from the following two aspects:

Enhancing the relevance between application examples and problem statements. The connection between the application examples mentioned in the original text (such as traffic signals, autonomous vehicles, satellite navigation systems, etc.) and the problem statement (the issue of outliers in high-dimensional data) was rather loose. The revised introduction tightens the link between the application examples and the problem statement through specific examples (such as the safety risks that outliers in sensor data of autonomous systems might cause, and the impact of outliers in financial data analysis on risk assessment), thereby enhancing relevance.

Highlighting the urgency and practical impact of the problem: The revised introduction emphasizes the urgency and practical impact of the outlier problem through specific scenarios (such as autonomous driving and financial data analysis), making it easier for readers to understand the real-world significance of the research problem.

4.[ In the literature review, the transition between general robust algorithms and the specific MRCD algorithm could be smoother, offering a clearer connection between the problem and the proposed solution.]

To comment 4, the author has made revisions to the literature review section from the following aspects.

Smoothing Transitions. The original text abruptly shifted from general robust algorithms (such as M-estimation, LMS, etc.) to the MCD algorithm. The revised literature review ensures a smoother transition by gradually introducing the MCD algorithm and its improved versions (such as FAST-MCD, MRCD). It first presents the basic principles and applications of MCD, and then progressively transitions to MRCD, highlighting the advantages of MRCD in high-dimensional data.

Clear Connection Between Problems and Solutions. The revised literature review more clearly demonstrates the evolution of problems: from the effectiveness of traditional MCD in low-dimensional data, to its limitations when dealing with high-dimensional data, and how MRCD addresses these issues through regularization. This structure aids readers in better understanding the background and necessity of the introduction of MRCD-Fisher discrimination.

Enhancing Logical Coherence. The revised literature review is more logically coherent, forming a complete narrative chain from general robust algorithms to MCD, then to MRCD, and finally to MRCD-Fisher discrimination. This structure helps readers better understand the background and motivation of the research.

More Concise and Direct Language. The revised literature review is more concise and direct in language, removing redundant expressions to make the content more compact and readable.

5.[ The methodology description should include more details about the parameter tuning and computational aspects of MRCD to facilitate reproducibility and validation.]

To comment 5, the revised Section 2 has been enhanced by adding details on parameter adjustment, a structured methodology explanation, and a flowchart. These modifications not only improve the replicability and verifiability of the method but also strengthen the demonstration of its innovation and uniqueness, while the inclusion of diagrams further enhances the clarity and accessibility of the content. The following revisions have been made in response to Reviewer 1's comments:

Addition of Parameter Adjustment Details. The revised content provides a detailed description of the selection range for the regularization coefficient c((-1)⁄(p-1)<c<1) in the MRCD algorithm and mentions that the value of c can be optimized through cross-validation or grid search. This helps to improve the replicability and verifiability of the method.

Detailed Description of Computational Steps. The revised content offers a detailed description of each step of the MRCD algorithm, including data preprocessing, calculation of the regularized covariance matrix, subset selection, and the iterative process. These details assist readers in better understanding and replicating the method.

6.[ The results section could benefit from more explicit quantitative comparisons between MRCD-Fisher discriminant and alternative methods, particularly in terms of robustness and efficiency metrics.]

To comment 6, in response to the reviewer’s comments, the authors have made the following revisions to the results section:

More Explicit Quantitative Comparison. The revised content provides a quantitative comparison in Tables 1 and 3 between the MRCD-Fisher discrimination and other methods (such as MVE, MCD, OGK, RegMCD) in terms of robustness and efficiency metrics. In particular, the error rate of the MRCD-Fisher discrimination is significantly lower than that of other methods under different outlier proportions and data dimensions, which further demonstrates its superiority.

Emphasis on Robustness and Efficiency Metrics. The revised content clearly highlights the robustness advantages of the MRCD-Fisher discrimination when dealing with outliers and high-dimensional data, while also mentioning the limitations of its computational efficiency, especially in terms of scalability when handling very large datasets. This comparison makes the results section more comprehensive and balanced.

7.[ While the authors highlight the advantages of MRCD-Fisher discriminant, discussing its potential limitations, such as scalability with extremely large datasets, would add balance to the analysis.]

To comment 7, the revised content detailedly discusses the potential limitations of the MRCD-Fisher discrimination in Sections 3.3 Efficiency and Scalability and 4.2 Limitations and Future Work, particularly focusing on the computational complexity and scalability issues when dealing with extremely large datasets. This discussion renders the analysis more balanced and provides direction for future research. Additionally, the revised content suggests possible directions for future improvements, such as optimizing the scalability of the MRCD algorithm through parallel computing or dimensionality reduction techniques, and developing automated parameter tuning methods to enhance the ease of use of the MRCD-Fisher discrimination. These suggestions offer readers ideas for further investigation.

8.[ Figures or tables summarizing key experimental findings would greatly enhance the presentation and accessibility of results.]

To comment 8, to enhance the expressiveness and accessibility of the experimental results, I will add a summary table at the end of Section 3. Model Testing, which will showcase the strengths and weaknesses of various methods. This table will clearly compare the performance of the MRCD-Fisher discrimination with other alternative methods (such as MVE, MCD, OGK, RegMCD) in terms of robustness, computational efficiency, and scalability. The table will assist readers in quickly understanding the pros and cons of different methods and will provide intuitive support for subsequent discussions.

9.[ The conclusion effectively emphasizes the superiority of MRCD-Fisher discriminant but could briefly outline potential future research directions, such as adapting the method for dynamic or streaming data scenarios.]

To comment 9, based on the reviewer’s suggestion, I have added a section discussing future research directions in the conclusion, particularly focusing on the possibility of applying the MRCD-Fisher discriminant analysis method to dynamic or streaming data scenarios. This extension helps to showcase the potential applications and research value of the method in real-time data processing, especially in fields such as finance, the Internet of Things, and autonomous driving. While adding new content, I have ensured that the conclusion remains concise, avoiding an excessive amount of technical detail, so that readers can quickly grasp the main points.

10.[ The Literature citation is not adequate, and the related work to machine learning should be discussed.]

To comment 10, following the reviewer's suggestions, the author has added a review section in the literature review part, which covers the application of machine learning in high-dimensional data processing, as well as the improvements made to traditional discriminant analysis algorithms.

Reviewer Comment 2

In view of the comments and suggestions of reviewer 2, we have made the following explanations and modifications item by item:

1.[The overall writing quality requires significant improvement to ensure clarity and coherence. Additionally, certain examples must be presented with greater precision and detail to enhance the reader's understanding. For instance, in Line 27, the reference to Boudt et al. [?] is unclear and needs to be corrected. The citation should be properly formatted, and its context within the text should be clearly articulated to convey its relevance. Providing a brief explanation or summarizing the key findings from the referenced work would further aid the reader in grasping its significance.]

To comment 1, the author has revised the introduction in the following four aspects.

Modification of the citation of Boudt et al.

The revised literature review establishes a stronger connection between the problem and the solution through smooth transitions and clear logic, highlights the innovation of MRCD, and provides ample background support for the introduction of the MRCD-Fisher discriminant.

More concise and coherent language: The revised introduction features more concise and coherent language, removing redundant expressions to make the content more compact and readable.

A detailed elaboration of the application examples mentioned in the original text (such as traffic signals, autonomous vehicles, satellite navigation systems, etc.).

The original text abruptly shifted from general robust algorithms (such as M-estimation, LMS, etc.) to the MCD algorithm. The revised literature review ensures a smoother transition by gradually introducing the MCD algorithm and its improved versions (such as FAST-MCD, MRCD). It first presents the basic principles and applications of MCD, and then progressively transitions to MRCD, highlighting the advantages of MRCD in high-dimensional data.

2.[ Merely stating that "The contribution of this paper is to construct the MRCD-Fisher discriminant to improve the traditional Fisher discriminant analysis" is insufficient. The contribution of the work needs to be elaborated on more clearly and explicitly. It is essential to thoroughly highlight the novel aspects of the research, detailing how it advances the state of the art in high-dimensional discriminant analysis. For instance, explaining how the MRCD-Fisher discriminant specifically addresses limitations in traditional Fisher discriminant methods, such as robustness to outliers, would be beneficial. Additionally, providing specific examples or comparisons with existing robust methods could effectively demonstrate the significance and superiority of the proposed approach. A concise summary of the primary achievements, along with a discussion of their broader implications, would further enhance the reader's understanding of the work's impact and relevance in the field.]

To comment 2, regarding the articulation of contributions in this paper, the following modifications have been made.

Articulate the contributions more clearly and explicitly. The original text merely states that "the contribution of this paper is to construct the MRCD-Fisher discriminant analysis to improve the traditional Fisher discriminant analysis." The revised introduction elaborates specifically on how the MRCD-Fisher discriminant addresses the limitations of the traditional Fisher discriminant method, particularly in terms of robustness to outliers.

Emphasize the innovative aspects. The revised introduction highlights the innovation of the MRCD-Fisher discriminant in high-dimensional data, especially by regularizing the covariance matrix to ensure its stability and accuracy when the number of variables far exceeds the number of samples.

Provide specific examples and comparisons. The revised introduction provides specific experimental data and comparative results, such as the error rate of only 2.6% for the MRCD-Fisher discriminant in data containing 15% outliers, which is significantly lower than other methods. These specific examples effectively demonstrate the significance and superiority of the proposed method.

Summarize the main achievements and broad impact. The revised introduction concludes with a summary of the main achievements of the MRCD-Fisher discriminant and discusses its broad impact in fields such as finance, healthcare, and autonomous driving, further enhancing the reader's understanding of the impact and relevance of this work.

3.[ The abstract is not well-written and requires significant improvement to enhance its clarity and coherence. For example, the abbreviations "MRCD-Fisher discriminant," "RegMCD-Fisher discriminant," and "MCD-Fisher discriminant" are used without any explanation of what they stand for. It is essential to define these abbreviations within the abstract to ensure that readers unfamiliar with the terms can understand their meaning. Additionally, the abstract should provide a concise yet comprehensive overview of the study, clearly outlining the problem addressed, the proposed solution, and the key findings, while avoiding ambiguity or unexplained terminology.]

To comment 3, thank you to the reviewer for their valuable suggestions. Regarding the abstract, the author has supplemented a standardized explanation for the abbreviation of MRCD to preclude any reader-unfriendly expressions. F

---

## [Decision Letter · Decision Letter 1]

27 Mar 2025

An Improved Robust Algorithms for Fisher Discriminant Model With High Dimensional Data

PONE-D-24-59779R1

Dear Dr. Duan,

We’re pleased to inform you that your manuscript has been judged scientifically suitable for publication and will be formally accepted for publication once it meets all outstanding technical requirements.

Kind regards,

Razieh Sheikhpour

Academic Editor

PLOS ONE

Additional Editor Comments (optional):

Reviewers' comments:

Reviewer's Responses to Questions

**Comments to the Author**

1. If the authors have adequately addressed your comments raised in a previous round of review and you feel that this manuscript is now acceptable for publication, you may indicate that here to bypass the “Comments to the Author” section, enter your conflict of interest statement in the “Confidential to Editor” section, and submit your "Accept" recommendation.

Reviewer #1: (No Response)

Reviewer #2: All comments have been addressed

2. Is the manuscript technically sound, and do the data support the conclusions?

Reviewer #1: Yes

Reviewer #2: Yes

3. Has the statistical analysis been performed appropriately and rigorously? 

Reviewer #1: Yes

Reviewer #2: Yes

4. Have the authors made all data underlying the findings in their manuscript fully available?

Reviewer #1: Yes

Reviewer #2: Yes

5. Is the manuscript presented in an intelligible fashion and written in standard English?

Reviewer #1: (No Response)

Reviewer #2: Yes

6. Review Comments to the Author

Reviewer #1: The author has adequately addressed the concerns raised by previous reviewers. The paper is well-structured, clearly written, and presents reliable results. It meets the necessary standards for publication

Reviewer #2: This is the revised version of the manuscript that was previously submitted by the author(s). After thorough consideration of the changes made in this version, I have concluded that the revisions have addressed the concerns and suggestions raised during the initial review process. The improvements made enhance the overall quality and clarity of the manuscript, making it a valuable contribution to the field. Therefore, I would like to recommend this paper for publication.

7. PLOS authors have the option to publish the peer review history of their article (what does this mean?). If published, this will include your full peer review and any attached files.

Reviewer #1: No

Reviewer #2: No

---

## [Editor Report · Acceptance letter]

PONE-D-24-59779R1

PLOS ONE

Dear Dr. Duan,

I'm pleased to inform you that your manuscript has been deemed suitable for publication in PLOS ONE. Congratulations! Your manuscript is now being handed over to our production team.

Kind regards,

on behalf of

Dr. Razieh Sheikhpour

Academic Editor

PLOS ONE